# Synthesis of Hierarchical Titanium Silicalite-1 Using a Carbon-Silica-Titania Composite from Xerogel Mild Carbonization

Xinya Pei [1], Xiaoxue Liu [2], Xiaoyu Liu [1], Jinling Shan [1], Hui Fu [1], Ying Xie [1], Xuemin Yan [1], Xianzhu Meng [1], Yancheng Zheng [1], Geng Li [1], Qi Wang [1] and Hao Li [1,*]

[1]  College of Chemistry and Environmental Engineering, Yangtze University, Jingzhou 434023, China
[2]  College of Agriculture, Yangtze University, Jingzhou 434000, China
*  Correspondence: haoli@yangtzeu.edu.cn; Tel.: +86-716-8060933; Fax: +86-716-8060650

**Abstract:** Hierarchical titanium silicalite-1 (HTS-1) zeolites are an important class of catalytic materials due to their enhanced mass transfer and improved catalytic performance. In this study, HTS-1 zeolites have been successfully prepared by the hydrothermal crystallization of carbon-silica-titania (CST) composites. Compared with the direct carbonization method, the mild carbonization of $SiO_2$-$TiO_2$/Tween 40 xerogel in the presence of sulfuric acid can effectively improve both the content and mesoporous structure of carbon material in the CST composites, which enables carbon materials to better play the role of a mesoporous template during the crystallization process. The resultant zeolite has both ordered micropores and interconnected mesopores and macropores, which are similar to the skeleton of the carbon template trapped in the TS–1 crystals. Moreover, the HTS–1 zeolite displays outstanding catalytic performance in oxidative desulfurization of bulky sulfur compounds.

**Keywords:** hierarchical TS-1; Tween 40; sulfuric acid pre-treatment; carbon-silica-titania composites; oxidative desulfurization

---

## 1. Introduction

In recent decades, Titanium silicalite-1 (TS-1) has attracted much attention due to its interesting catalytic performances in selective oxidation reactions, such as hydroxylation of aromatics, epoxidation of olefins, and especially the oxidation of S-containing organic compounds [1,2]. However, its intrinsic microspores hinder the diffusion of bulky organic sulfide molecules to active sites, which restricts its further applications in the oxidative desulfurization process [2]. To overcome this drawback, intensive research on the preparation of TS-1 zeolites with hierarchical pores have been carried out [3–10].

Generally, there are two prominent strategies for the preparation of hierarchical TS-1 (HTS-1) zeolites: Top-down and bottom-up approaches [3]. The top-down approach is the destructive desilication of TS-1 zeolites through post–synthesis treatments in alkali [11] or acid solution [3]. However, these treatments are aggressive and often induce loss of zeolite structure. In the bottom-up approach, the preparation of HTS-1 zeolites is accomplished by using various soft or hard templates. The soft templates usually include traditional surfactants [12], organosilane agents [13], cationic polymers [4], and copolymers containing quaternary ammonium groups [5]. The hard templates mainly include carbon black [14], carbon nanotube [15], and mesoporous carbon [6]. However, their main disadvantage is that a large amount of sometimes complex and expensive organic materials must be sacrificed. Soft templates are generally easy to disperse throughout the synthesis system, whereas hard templates are difficult to disperse in the synthesis system due to the weak interaction between templates and inorganic precursors. To solve this problem, polyvinyl alcohol, which possesses rich –OH groups, has been used as a linking agent to connect multi-wall carbon nanotubes and TS-1

precursors [15]. Unfortunately, the mesopore volume of HTS-1 does not increase much compared to TS-1. Recently, by carbonization of sucrose onto silica and titania raw materials, the carbon template is easy to be enwrapped in the crystallization process [7]. However, since there are only physical mixing and dispersion between the sucrose and inorganic species, the improvement may be poor.

Recently, Tween surfactants have been widely used to synthesize mesostructured silica and zirconia, and to tune the morphology and pore structure of hierarchical meso-macroporous silica [16–19]. The special micelles aggregated by Tween surfactants can greatly increase the interactions between their hydrophilic shells and inorganic Si and Ti species [16,17,20]. Moreover, using sulfuric acid as the carbonization catalyst, carbon or silica–carbon materials can be easily prepared through a mild carbonization process [21,22]. The action of sulfuric acid is carried out by dehydration and sulfonation reactions, promoting the formation of aromatic structures and facilitating the crosslinking process [22]. Here, we synthesized HTS–1 zeolites by the hydrothermal crystallization of carbon-silica-titania (CST) composites, using tetrapropylammonium hydroxide (TPAOH) as the microporous template. The uniformly hybrid $SiO_2$-$TiO_2$/Tween 40 xerogel (denoted as ST/T-40) was prepared by the sol-gel method in the presence of Tween 40 (T-40). The effect of two different patterns for xerogel carbonization, i.e., direct carbonization and mild carbonization with the aid of sulfuric acid, on the pore structure of HTS–1 zeolites is demonstrated. Moreover, catalytic properties of the resulting samples in the oxidative desulfurization of bulky sulfur compounds are studied.

## 2. Results and Discussion

### 2.1. Structural Characterizations

The ST/T-40 xerogel is proven to have a mesostructure by small-angle XRD, and its mesostructure changes little upon sulfuric acid pre-treatment (Figure S1a). But their mesostructures collapse upon carbonization, as evidenced by the disappearance of the diffraction peak (Figure S1b). However, the $SiO_2$-$TiO_2$ composites obtained by combustion of corresponding parent materials possess a mesostructure (Figure S2). The adsorption peaks for C–H stretching and bending vibrations (2800–3000 $cm^{-1}$, 1464 $cm^{-1}$, and 1350 $cm^{-1}$) and C=O stretching vibration (1728 $cm^{-1}$) in the FT-IR spectra (Figure S3) reveal the presence of T-40 in the ST/T-40 xerogel [22,23]. After the sulfuric acid pre–treatment, the intensity of these peaks decreases sharply. However, these characteristic peaks of T-40 are no longer observed upon carbonization, indicating that T-40 in the xerogel and CSTp (CST composite prepared by pre-treating the xerogel with dilute sulfuric acid) are converted into carbon materials [22]. Moreover, the presence of Si–O–Ti linkages in the xerogel and CST composites is identified by the absorption band at 960 $cm^{-1}$ (Figure S3), which can make Ti more dispersive and stable, and avoid the precipitation of extra-framework titanium dioxide [24].

The carbon contents of CSTd (CST composite prepared by direct carbonization of ST/T-40 xerogel) and CSTs (CST composite prepared by carbonization of ST/T-40 xerogel with the aid of sulfuric acid) obtained by calcination in air at 823 K were 19.4% and 38.9%, respectively. The results suggest that it is important to pretreat the xerogel with $H_2SO_4$ to obtain a higher carbon content in the resulting CST composite. The carbon materials from CSTs and as-synthesized HTS-1s possess considerably higher mesoporous volume, BET surface area, and external surface area than those from the CSTd and as-synthesized HTS-1d (Table S1). The pore size distributions of carbon materials from the CSTs and as-synthesized HTS–1s are both in the range of 2–95 nm (centered at 16 nm), which changes little upon the hydrothermal crystallization (Figure S4). These results indicate that the mild carbonization of the xerogel with sulfuric acid can effectively improve both the content and mesoporous structure of carbon materials in the CST composites, which are favourable for carbon materials to better play the role of mesoporous template.

Figure S5 shows the XRD patterns of TS-1, HTS-1d, and HTS-1s. All samples display the characteristic of MFI topology without the impure phase. Figure S6 shows the FT-IR spectra of samples. The band at 960 $cm^{-1}$ is assigned to the stretching vibration of the Si–O–Ti bond or Si–O bond

perturbed by the framework Ti atoms [24]. The band at 550 cm$^{-1}$ is attributed to the characteristic of MFI topology [25]. UV–Vis spectra of samples show a strong absorption band at 210 nm, which is attributed to the isolated Ti (IV) species (Figure S7). A weak absorption of the samples at about 330 nm is observed, indicating the existence of some non-framework titanium dioxide species [8,9].

Figure 1 and Figure S8 show the nitrogen adsorption–desorption isotherms and pore size distributions of samples. HTS-1d shows a typical type-I isotherm as with TS-1, while HTS-1s exhibits the type-IV isotherms. A small hysteresis loop in the relative pressure range of $0.50 < P/P_0 < 0.85$ is observed for HTS-1d, and a large hysteresis loop at $P/P_0 > 0.7$ is obtained for HTS-1s, revealing the presence of mesopores or macropores. In contrast to the HTS-1d showing a broad pore size distribution (3–77 nm), the pore size of HTS-1s is concentrically distributed at ~13 nm (5–95 nm) and the pore volume is much higher (Figure 1b). Compared with TS-1 and HTS-1d, the mesoporous volume, BET surface area, and external surface area of HTS-1s are improved remarkably by introducing the sulfuric acid pre-treatment in the synthesis process (Table 1).

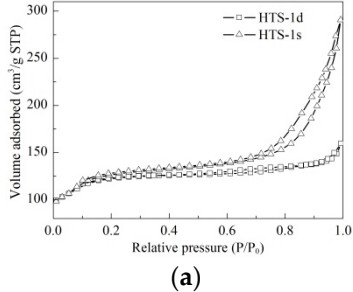 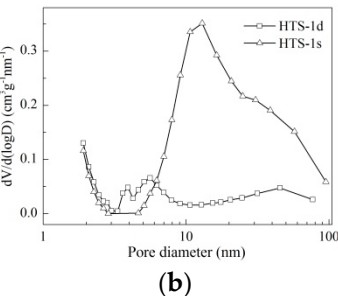

(**a**) (**b**)

**Figure 1.** N$_2$ adsorption-desorption isotherms (**a**) and pore size distributions (**b**) of hierarchical titanium silicalite-1 (HTS-1) samples.

**Table 1.** Compositions and textural properties of titanium silicalite-1 (TS-1) and HTS-1 zeolites.

| Sample | Si/Ti [a] | S$_{BET}$ (m$^2$ g$^{-1}$) | S$_{micro}$ (m$^2$ g$^{-1}$) | S$_{ext}$ (m$^2$ g$^{-1}$) | V$_{micro}$ (cm$^3$ g$^{-1}$) | V$_{meso}$ (cm$^3$ g$^{-1}$) |
|---|---|---|---|---|---|---|
| TS-1 | 32.4 | 400.0 | 280.7 | 119.3 | 0.14 | 0.03 |
| HTS-1d | 33.1 | 387.8 | 258.0 | 129.8 | 0.13 | 0.07 |
| HTS-1s | 33.9 | 409.0 | 244.5 | 164.5 | 0.12 | 0.29 |

[a] Measured by ICP-OES.

Figure 2 and Figure S9 reveal the SEM images of CST composites, carbon materials, TS-1, and HTS-1 zeolites. Silica and titanium particles are well enclosed by carbon materials, and plenty of 5–20 nm mesopores are observed for CSTs (Figure S9a,b). The carbon material from CSTd only shows a lot of shallow grooves on the surface, while the carbon material from CSTs shows a three-dimensional continuously mesoporous structure with a pore wall of 5–80 nm (Figure 2a,b). Upon hydrothermal synthesis, the pore structure of carbon material from HTS-1s changes little, while the pore structure of carbon material from HTS-1d is improved to some extent (Figure S9c,d, Figure S4, and Table S1). TS-1 shows a smooth crystal surface and twinned coffin morphology (Figure S9e), while HTS-1 samples have a rough surface and non-uniform morphology (Figure 2c,d). HTS-1s crystals are an aggregated structure composed of smaller nanoparticles (10–50 nm) and the mesopores between crystals are connected to each other and to the outer surface of the zeolite (Figure 2c). However, the HTS-1d particles only show a rough surface with some shallow grooves on the zeolite surface (Figure 2d). These results of HTS-1s are consistent with those of the framework of carbon materials from CSTs and as-synthesized HTS-1s, which are also three dimensionally interconnected mesoporous networks.

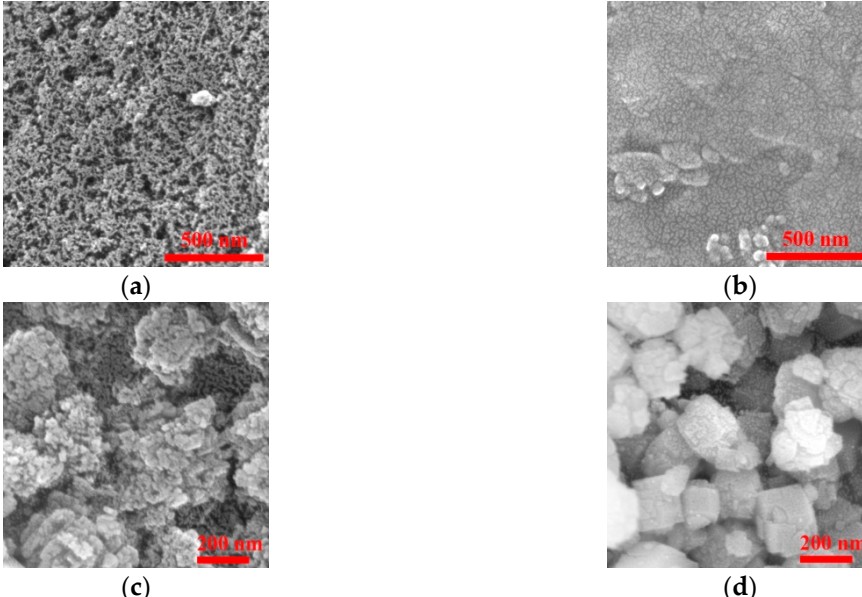

(**a**)

(**b**)

(**c**)

(**d**)

**Figure 2.** SEM images of carbon materials obtained by removals of silica and titania from CSTs (**a**) and CSTd (**b**) with hydrofluoric acid aqueous solution, HTS-1s (**c**), and HTS-1d (**d**).

The TEM images of HTS-1s (Figure 3) provide direct evidence for the existence of a lot of mesopores and macropores in the zeolite. Both intracrystalline and intercrystalline voids are observed in HTS-1s, and part of them link directly to the surface of crystals. Moreover, the lattice lines of MFI topology with a micropore size of ~0.5 nm can be unambiguously identified (Figure 3b). The results of SEM and TEM analyses show that the ordered micropores of TS-1 crystals are partially connected to the mesopores, thereby forming the required interconnected hierarchical pore system. Therefore, it is obvious that TS-1 nanoparticles are well encapsulated by carbon templates during the crystallization process.

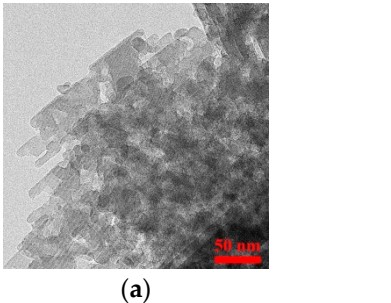

(**a**)

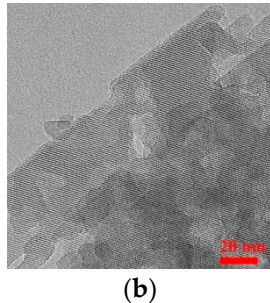

(**b**)

**Figure 3.** TEM images taken at low (**a**) and high (**b**) magnification of HTS-1s.

### 2.2. Catalytic Performance on Oxidative Desulfurization

The catalytic properties of samples are studied by the oxidation of Dibenzothiophene (DBT) and 4,6-dimethyldibenzothiophene (4,6-DMDBT), using Tert–butyl hydroperoxide (TBHP) as the oxidatant. FT-IR spectra of the oxidized products demonstrate that DBT and 4,6-DMDBT are oxidized into dibenzothiophene sulfone and 4,6-dimethyldibenzothiophene sulfone, respectively (Figure S10) [26]. The reaction pathways are shown in Scheme S1. Sulfur compounds are first oxidized into sulfoxides, which are very unstable and are oxidized fast into sulfones [9,10]. Furthermore, it is worth noting that the oxidation products (sulfones) are insoluble and precipitate from the reaction medium, and catalysts can also precipitate from the reaction medium immediately after the reaction is finished (Figure S11). These interesting phenomena can facilitate their removal from model fuels by simple centrifugation or filtration.

As shown in Figure 4, TS-1 gives a DBT conversion of 67.8% after 3 h and a 4,6-DMDBT conversion of 71.9% after 2 h, while the conversions of DBT and 4,6-DMDBT for HTS-1d can reach 93.8% and 92.4%, respectively. Substantially improved catalytic activity is demonstrated by HTS-1s, which converts over 99.5% of both DBT and 4,6-DMDBT. The similar titanium contents of TS-1 and HTS-1 samples (Table 1) suggest that the enhanced activity demonstrated by HTS-1 samples is most probably due to the improved accessibility of DBT and 4,6-DMDBT to the active sites [8]. Compared with TS-1 and HTS-1d, the external surface area and mesopore volume of HTS-1s are much higher (Table 1), which can effectively overcome the diffusion limitation for bulky sulfur compounds. Meanwhile, HTS-1s crystals are an aggregated structure composed of smaller nanoparticles (10–50 nm), which can greatly shorten the diffusion path of reactant molecules [6]. In addition, HTS-1s can be easily recycled (Figure S12 and Figure 5). When HTS-1s was recovered by simple centrifugation and drying, the conversions of DBT and 4,6-DMDBT began to clearly decrease after the fifth run. The deactivation is mainly caused by the blockage of active sites by reaction products (Figure S13) [3]. However, the used catalyst can be fully regenerated through combustion at 823 K for 2 h (Figure 5 and Figure S13). After the 15th run, HTS-1s still keeps high conversions of DBT (99.5%) and 4,6-DMDBT (99.4%) without deactivation. The regenerated catalysts were further characterized by XRD and SEM techniques (Figures S14 and S15). The results indicate that HTS-1s can withstand repeated calcinations at 823 K, and the pore structure remains good. The good reusability suggests that HTS-1s is a good catalyst for deep oxidation desulfurization.

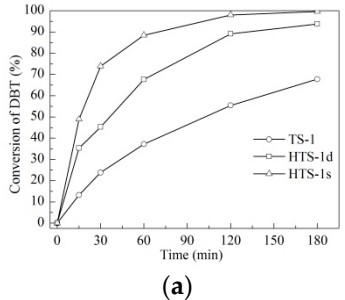　　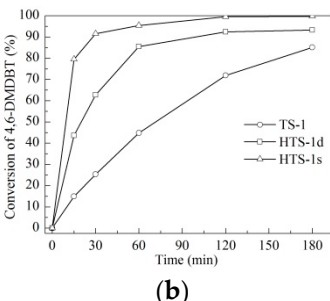

(**a**)　　　　　　　　　　　　　　　　　　　　　(**b**)

**Figure 4.** Catalytic oxidation of dibenzothiophene (DBT) (**a**) and 4,6-dimethyldibenzothiophene (4,6-DMDBT) (**b**) over TS-1, HTS-1d, and HTS-1s.

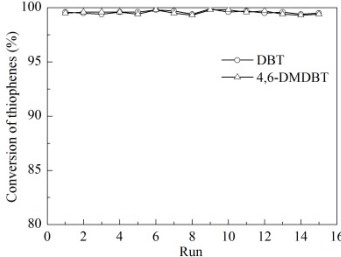

**Figure 5.** Recycle tests for the catalytic oxidation of DBT and 4,6-DMDBT over regenerated HTS-1s. In each run, a small amount of fresh catalyst (<5%) was added in order to make up for the loss during the recovery process.

## 3. Experimental

### 3.1. Synthesis

#### 3.1.1. Synthesis of Hybrid SiO$_2$-TiO$_2$/T-40 Xerogel

The hybrid SiO$_2$-TiO$_2$/T-40 xerogel (denoted as ST/T-40) was prepared by two step sol-gel processes in the presence of T-40 [27]. Typically, 33.3 g of tetraethylorthosilicate (Aladdin) was hydrolyzed at room temperature with a 0.05 mol/L HCl solution (11.5 g). After the clear solution was cooled to 273 K,

1.8 g of tetrabutylorthotitanate (Aldrich) dissolved in 10.6 g of isopropyl alcohol was added dropwise. Then, 26.7 g of T-40 dissolved in 60 g of deionized water was added. After stirring at 313 K for 2 h, 1.1 g of TPAOH (40%, Alfa) was added dropwise and, finally, a pale yellow gel was formed. The molar composition of gel was: $SiO_2 = 1$, $TiO_2 = 0.033$, T-40 = 0.13, $H_2O = 25$ and TPAOH = 0.014. Finally, the gel was dried fully at 353 K and grinded into fine powders. For comparison, $SiO_2$-$TiO_2$ (denoted as ST) xerogel was prepared in a similar manner without the presence of T-40.

### 3.1.2. Synthesis of Carbon-Silica-Titania (CST) Composites

The CST composites were prepared by carbonization of hybrid ST/T-40 xerogels with [22] or without the aid of sulfuric acid. Typically, 10.0 g of xerogel was dispersed in an aqueous solution of sulfuric acid (100 mL of deionized water and 2.0 mL of 98% $H_2SO_4$) under stirring for 1 h. This mixture was dried completely at 373 K, and then heated at 433 K for ~12 h to obtain a black solid, which was grinded into fine powders (denoted as CSTp). Finally, CSTp powders were heated up to 773 K (1 K/min) and maintained for 12 h under a $N_2$ flow. The resulting CST composite was denoted as CSTs, where s represents the sulfuric acid pre–treatment. For the direct carbonization method, the CST composite (denoted as CSTd) was synthesized in a similar manner without the aid of sulfuric acid.

### 3.1.3. Synthesis of HTS-1 Zeolites

In a typical synthesis, the CST composites were impregnated with an aqueous solution of TPAOH for 4 h under stirring. The molar composition of gel was: $SiO_2 = 1$, $TiO_2 = 0.033$, TPAOH = 0.45 and $H_2O = 16.6$. The gel was transferred to a Teflon-lined autoclave, and crystallized under autogeneous pressure at 443 K for 24 h. The resulting black solid was washed, dried, and calcined at 823 K for 5 h. The HTS-1 zeolites produced from CSTs and CSTd were denoted as HTS-1s and HTS-1d, respectively. Moreover, TS-1 was prepared through hydrothermal crystallization of the ST xerogel in a similar manner.

### 3.2. Characterization

The samples were characterized by XRD patterns (Panalytical Empyrean, Cu-K$\alpha$ radiation), FT-IR spectrometer (Nicolet 6700, KBr pellet technique), UV–Vis spectroscopy (PerkinElmer Lambda 650S), SEM images (TESCAN MIRA3), TEM images (JEOL JEM-2100F), and ICP-OES analysis (PerkinElmer Optima 8000). $N_2$ adsorption-desorption isotherms were obtained by a Micromeritics ASAP 2020HD88 analyzer at 77 K. Prior to analysis, the samples were degassed at 573 K for 4 h. The specific surface area ($S_{BET}$) was evaluated using a BET equation, whereas the micropore volume ($V_{micro}$), external surface area ($S_{ext}$), and micropore area ($S_{micro}$) were calculated by t-plot method. The total pore volume was measured from the amount adsorbed at a relative pressure of 0.99. Moreover, the cumulative pore volume ($V_{meso}$) and pore size distribution were calculated from the desorption branch of isotherm by the Barrett–Joyner–Halenda (BJH) model.

### 3.3. Catalytic Activity

The oxidative desulfurization of dibenzothiophene (DBT) and 4,6-dimethyldibenzothiophene (4,6-DMDBT) were carried out in a 50 mL two-neck glass flask immersed in an oil bath and equipped with a condenser. DBT and 4,6-DMDBT were dissolved in n-octane to prepare model fuels. The concentrations of DBT and 4,6-DMDBT were 1000 and 500 ppmw, respectively. Typically, 50.0 mg catalyst and 10.0 mL model fuel were mixed in the flask and heated at 333 K under stirring. Tert-butyl hydroperoxide (TBHP, Aldrich, ~5.5 mol/L in decane), with a TBHP/S molar ratio of 2, was then added to the mixture to start the reaction for the required time (3 h for DBT and 2 h for 4,6-DMDBT). Then, the reactive mixture was cooled and centrifuged. The oil phase was analyzed on a GC 126N chromatography, equipped with a flame photometric detector (FPD) and an HP-5 capillary column (0.25 μm × 0.32 mm × 30 m). The residual DBT and 4,6–DMDBT was quantified by the peak area normalization method. The conversion of thiophenes (DBT and 4,6-DMDBT) (X) was defined

as the amount of sulfide reacted divided by the initial amount of sulfide added. The products and catalysts were separated using acetonitrile as a solvent.

## 4. Conclusions

In summary, a hierarchical TS-1 zeolite was successfully prepared using TPAOH as a microporous template and carbon material from mild carbonization of xerogel as a mesoporous template. By pre-treating the xerogel with dilute sulfuric acid, the content and mesoporous structure of carbon material are improved effectively, thereby forming the secondary mesopores and macropores in the TS-1 zeolite after calcination. Compared with TS-1 and HTS-1d, HTS-1s shows excellent catalytic activity and stability in the oxidative desulfurization of DBT and 4,6-DMDBT. This strategy can be applied to synthesize a series of hierarchical Ti-containing zeolites with special properties.

**Supplementary Materials:** The following are available online at http://www.mdpi.com/2073-4344/9/8/672/s1, Figure S1. XRD patterns of samples: (a) ST/T-40 xerogel and CST composite prepared by pre-treating the ST/T-40 xerogel with dilute sulfuric acid (CSTp); (b) CST composites prepared by carbonization of ST/T−40 xerogels with (CSTs) or without (CSTd) the aid of sulfuric acid–, Figure S2. XRD patterns of $SiO_2$–$TiO_2$ composites obtained by combustion of ST/T-40 xerogel (1), CSTd (2), and CSTs (3) in air at 823 K for 5 h, Figure S3. FT-IR spectra of ST/T-40 xerogel and CST composites: (1) ST/T-40 xerogel; (2) CSTp; (3) CSTs; (4) CSTd, Figure S4. Nitrogen adsorption-desorption isotherms (a) and pore size distributions (b) of carbon materials obtained by removals of silica and titania from different parent materials with hydrofluoric acid aqueous solution: (●) carbon material from CSTd; (■) carbon material from as-synthesized HTS−1d; (o) carbon material from CSTp; (□) carbon material from CSTs; (+) carbon material from as-synthesized HTS-1s, Figure S5. XRD patterns of TS-1, HTS-1d, and HTS-1s, Figure S6. FT-IR spectra of TS-1, HTS-1d, and HTS-1s, Figure S7. UV–Vis spectra of TS-1, HTS-1d, and HTS-1s, Figure S8. Nitrogen adsorption–desorption isotherms of TS-1, Figure S9. SEM images of CSTs (a), CSTd (b), carbon materials obtained by removals of silica and titania from as-synthesized HTS-1s (c) and HTS-1d (d) with hydrofluoric acid aqueous solution, and TS-1 (e), Figure S10. FT-IR spectra of reactants and their corresponding products: (a) DBT and its oxidation product; (b) 4,6-DMDBT and its oxidation product-, Figure S11. Image of a typical reaction medium after the reaction was finished for 3 min, Figure S12. Recycle tests in the oxidation of DBT and 4,6-DMDBT over HTS-1s recovered by centrifugation and drying at 373 K, Figure S13. UV–Vis spectra of the fresh, used, and regenerated HTS-1s for DBT (a) and 4,6-DMDBT (b) oxidations: (1) HTS-1s regenerated by calcination after the 15th run; (2) HTS-1s recovered by centrifugation and drying after the sixth run; (3) product of DBT oxidation; (4) fresh HTS-1s; (5) HTS-1s regenerated by calcination after the 15th run; (6) HTS-1s recovered by centrifugation and drying after the sixth run; (7) product of 4,6-DMDBT oxidation, Figure S14. XRD patterns of fresh HTS-1s (1) and regenerated HTS−1s catalysts for DBT (2) and 4,6-DMDBT (3) oxidations, Figure S15. SEM images of the regenerated HTS-1s after the 15th run: (a) and (b) for 4,6-DMDBT oxidation; (c) and (d) for DBT oxidation, Table S1. Textural properties of carbon materials from different parent materials, Scheme S1. The reaction pathway for oxidative desulfurization of DBT and 4,6-DMDBT over HTS−1s.

**Author Contributions:** H.L. designed the experiments, analyzed the experimental data, and wrote the manuscript. X.P., X.L. (Xiaoxue Liu), X.L. (Xiaoyu Liu), and J.S. prepared the ST/T-40 xerogel, CST composites, and HTS-1 zeolites. H.F., Q.W., and Y.X. tested the catalytic performances of samples. X.P., X.M., and G.L. characterized the samples. H.L., Y.Z., and X.Y. revised the final version of the paper.

**Funding:** This research was funded by the National Natural Science Foundation of China (21303008) and Natural Science Foundation of Hubei Province of China (2012FFB00103).

**Conflicts of Interest:** The authors declare no conflict of interest.

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
