# Peer review of "Synthesis of Hierarchical Titanium Silicalite-1 Using a Carbon-Silica-Titania Composite from Xerogel Mild Carbonization"

_catalysts, doi:10.3390/catal9080672_

Round 1

Reviewer 1 Report

1.        All the figures should be modified. Scale bar, caption, contrast (Figure 2 (b))

2.        The difference in the catalytic activity of the catalysts should be clarified. Was the diffusion the main reason for low activity of typical TS-1? If so, the effect of the reaction time should be investigated.

3.        TBHP was used as oxidant. How about H2O2?

Reviewer 2 Report

The manuscript describes the preparation approach of  hierarchical TS-1 zeolite via carbon templated approach with using a Tween surfactant. The obtained sample demonstrated high catalytic performance in oxidative desulfurization reactions. The novelty of this work is to prepare nano-sized TS-1 crystals with a Tween surfactant as a carbon template. The manuscript seems to be well-organized and enough characterization were conducted so far. 

Followings are several minor comments.

Comment 1.

In the discussion about SEM images (page 3, line 115-), some figures are denoted as "carbon templates" (e.g., Fig.2a, 2b). What are these? Are they carbon residues after desilication of CST composite? And, the expression "dry-up bottom of a lake" is unclear. 

Comment 2.

In the catalytic test, what authors shows in Table 1, Figure 4, and Figure S12 should be denoted as "conversion of reactant thiophenes". I understand that the authors use the term "rate" in a meaning of "ratio of reacted DBT/DMDBT amount to initial amount". That is, however, confusing because "rate" generally means the "reaction rate" in the context of catalysis paper. Related expressions in a manuscript should also be corrected.

Comment 3. 

In the Scheme S1, the structure of 4,6-DMDBT is incorrect.
